# VELR: Efficient Video Reward Feedback via Ensemble Latent Reward Models

## Abstract

Reward feedback learning (ReFL) is effective for both text-to-image (T2I) and text-to-video (T2V) generation with image reward models (RMs). However, image RMs are misaligned with temporal objectives of T2V, motivating ReFL with video reward models. Nevertheless, directly deploying video RMs is impractical due to their large parameter size and the prohibitive cost of memory. To address this, we propose VELR: an efficient framework that employs ensemble latent reward models (LRMs) to predict rewards directly in latent space, bypassing expensive backpropagation through VAE decoders and video RMs. Specifically, we introduce the ensemble technique for the LRM, which enhances capacity, quantifies uncertainty, and mitigates reward hacking. VELR achieves a reduction of up to 150GB in memory, requiring as little as 12.4% of the memory compared to standard ReFL. Experiments on OpenSora, CogVideoX-1.5, and Wan-2.1 with large-scale video RMs demonstrate that VELR achieves comparable performance as standard ReFL and enables efficient and robust video RM-based ReFL at scales previously unattainable.

## 1 Introduction

Recent advances in text-to-video (T2V) generation have yielded increasingly powerful models (Hong et al., 2023; Zheng et al., 2024; Wan et al., 2025). However, these models may misalign with human preference and may produce undesirable, toxic, or harmful contents. Reinforcement learning with human feedback (RLHF) has emerged as a key technique to align generative models with human expectations, and has shown success in both language (Christiano et al., 2017; Ouyang et al., 2022) and T2I domains (Lee et al., 2023). In the context of T2V, several RLHF approaches have been explored, including policy optimization methods such as DDPO (Black et al., 2024), and GRPO (Xue et al., 2025; Liu et al., 2025a), as well as preference optimization methods such as DPO (Wallace et al., 2024; Wu et al., 2025b). While effective to some extent, these approaches are computationally expensive and often lead to only marginal improvements.

Reward feedback learning (ReFL) offers a more direct and effective alternative which guides the model's preferences by directly backpropagating reward signals (Xu et al., 2023). However, existing methods require maintaining gradients through both the VAE decoder and the reward model, leading to excessive memory overhead. Consequently, existing ReFL solutions mainly rely on image reward models (RMs) and require extensive memory optimizations to run (Yuan et al., 2024; Prabhudesai et al., 2025) which, despite improving perceptual quality, fail to capture temporal consistency critical for video generation. This mismatch motivates the use of video-based reward models to fine-tune diffusion models, and a recent work has demonstrated that fine-tuning with Vision-Language Models (VLMs) can significantly improve video generation models (Wu et al., 2025a). Yet, such gains come at the cost of extreme memory demands: recent high-performing video reward models (Liu et al., 2025b; Wang et al., 2025b) are typically built on large VLMs (Bai et al., 2025), making them prohibitively large and memory-intensive. As illustrated in Fig 1, even processing a single frame demands more than 100 GB of memory, which is infeasible for most hardware.

A promising direction is to bypass the decoder and reward model by training latent reward models (LRMs) that predict rewards directly from latent representations (Ding et al., 2025; Zhang et al., 2025). Ding first introduced the concept of LRM, but the implementation employed only a simple network architecture and was evaluated with image-based reward models on consistency models. While these explorations reveal that LRMs offer a promising direction for efficient ReFL, existing approaches fall short of providing reliable LRM solutions for large-scale video reward models. We

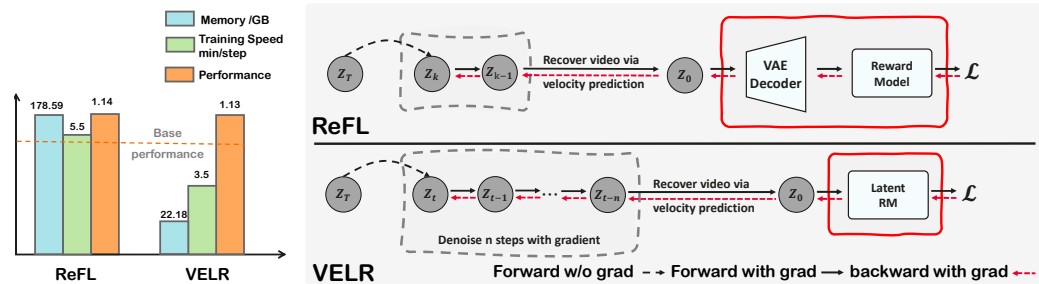

Figure 1: We propose **VELR**, a novel reward feedback learning framework that utilizes the ensemble latent reward model. Va novel reward feedback learning framework that leverages an ensemble latent reward model. Compared to ReFL, VELR substantially alleviates memory overhead, accelerates training, and maintains comparable visual fidelity. These improvements make it feasible to fine-tune T2V models with video-based reward models, a setting that was previously difficult to run.

find that the LRM proposed in (Ding et al., 2025) is insufficient to regress large-scale vRMs and exhibits poor generalization to out-of-distribution samples, leading to unreliable reward signals. This motivates the design of a more capable and robust LRMs tailored for ReFL in video generation.

To address this, we propose VELR: Efficient **V**ideo Reward Feedback via **E**nsemble **L**atent **R**eward Models. Our contributions are threefold:

(1) we introduce an ensemble LRM that enhances spatio-temporal capacity, provides uncertainty estimation, and mitigates reward hacking. This ensemble-LRM design significantly reduces memory consumption and accelerates training, thereby making ReFL feasible on sota T2V models.

(2) we carefully design a training procedure tailored for the ensemble-LRM ReFL framework with two key components. Specifically, the truncated mid-step setting enables fast and effective gradient updates, while the online alignment of LRMs ensures that the reward estimates remain calibrated to the ground-truth rewards.

(3) Combined, these techniques reduce memory usage by up to **87.6%** and make ReFL feasible for sota T2V models and vRMs, including OpenSora1.2 (Zheng et al., 2024), CogVideoX1.5(Hong et al., 2023; Yang et al., 2025), and Wan-2.1 (Wan et al., 2025) with video RMs up to 32B parameters. Extensive experiments demonstrate that VELR is efficient, scalable, and robust, enabling vRM-based ReFL at previously unreachable model and reward scales.

## 2 RELATED WORK

Reinforcement learning has become a standard paradigm for aligning diffusion-based video generation models with human preferences. Existing RLHF methods can be roughly categorized into three groups. Policy-gradient-based approaches, such as DDPO (Black et al., 2024), optimize the diffusion model as a stochastic policy but often suffer from instability and sample inefficiency. Group-based policy optimization methods, including DanceGRPO (Xue et al., 2025) and FlowGRPO (Liu et al., 2025a), estimate the advantage function via a group of samples, which substantially increases inference cost. Preference-optimization methods, such as DPO (Wallace et al., 2024; Zhang et al., 2024a; Wu et al., 2025b), construct paired samples to reflect human choices, but rely on large-scale curated preference datasets. Overall, while RLHF provides a principled alignment mechanism, these approaches are either unstable, computationally demanding, or data-intensive.

ReFL can be regarded as a variant of RLHF, which enables faster optimization and yields competitive performance (Xu et al., 2023). ReFL has already demonstrated strong performance in the image generation domain. Chen et al. (2024a) incorporated text-encoder feedback to strengthen semantic alignment, Fan et al. (2024) adapted ReFL for prompt tuning to enhance abstract concept understanding, Hyper-SD proposed a trajectory-segmented consistency model, integrating ReFL to accelerate image synthesis (Ren et al., 2024), Unifl unified multiple feedback signals to improve latent diffusion training (Zhang et al., 2024b), and ImageReFL applied ReFL to balance quality

and diversity in human-aligned diffusion models (Sorokin et al., 2025). A recent work Dollar explores the use of a latent reward model for ReFL (Ding et al., 2025). However, it focuses solely on fine-tuning consistency models with image-based reward models.

In the context of video generation, applying ReFL is challenging due to its high memory requirements and the complex design of the VAE decoder in T2V models (Hong et al., 2023). As a result, existing works have only explored ReFL with image-based reward models, such as InstructVideo(Yuan et al., 2024) and VADER (Prabhudesai et al., 2025). In contrast to these works, VELR is the first to apply the ReFL framework on T2V models using **large-scale video RMs**, and the latent reward model used in VELR significantly outperforms the one proposed by Dollar.

## 3 PRELIMINARIES

### 3.1 DIFFUSION MODEL

The diffusion models aim to approximate the data distribution $x_0 \sim q(x_0)$ by defining a stochastic process that gradually perturbs the data with Gaussian noise and then learns to reverse this process in order to recover samples from $q(x_0)$ (Rombach et al., 2022).

The forward process in the latent space is defined as a Markov chain, where a clean latent variable $z_0$ is progressively corrupted into $z_t$ through a predefined noise schedule $\alpha_t, t \in [1, \cdots, T]$ (Ho et al., 2020; Song et al., 2020):

$$z_t = \sqrt{\bar{\alpha}_t} z_0 + \sqrt{1 - \bar{\alpha}_t}, \epsilon, \quad \epsilon \sim \mathcal{N}(\mathbf{0}, \mathbf{I}), \tag{1}$$

with $\bar{\alpha}_t = \prod i = 1^t \alpha_i$ and $\epsilon \sim \mathcal{N}(\mathbf{0}, \mathbf{1})$ represents the normal Gaussian noise.

The reverse process seeks to recover $z_0$ from $z_t$ by learning a neural approximation of the injected noise. In the standard velocity parameterization, the model $v_\theta$ is trained to predict the ground-truth velocity $v_t$, leading to the following objective:

$$L_V(\theta) = \mathbb{E}_{x_0 \sim q(x_0), \epsilon \sim \mathcal{N}(0, I), t}[\omega_t || v_\theta(x_t, t) - v_t ||_2^2] \tag{2}$$

For rectified flow (Liu et al., 2022), the interpolation between clean data $x_0$ and noise $\epsilon$ is defined by coefficients $a_t = 1 - t, b_t = t$ with $t \in [0, 1]$. In this case, the velocity field becomes constant.

### 3.2 REWARD FEEDBACK LEARNING

ReFL is a reinforcement learning paradigm that leverages pretrained reward models to guide policy optimization. Given a reward model $R_p(v, c)$ trained to predict human or learned preferences over generated outputs, the latent diffusion model is considered as a policy $\pi_\theta(z_0|c)$. the ReFL updates the policy $\pi_\theta$ by maximizing the expected reward on a dataset $\mathcal{D}$ composed of prompts:

$$\theta^* = \arg\max_\theta \mathbb{E}_{c \sim \mathcal{D}, v \sim \pi_\theta(z_0|c)}[R_p(v, c)] \tag{3}$$

## 4 METHODOLOGY

To mitigate the excessive memory cost of pixel-space RMs while leveraging the superior capability of video RMs, we adopt a latent reward modeling strategy. Sec. 4.1 presents the architecture of the proposed latent reward model (LRM) and its training paradigm on a mixed dataset of real and generated videos. In Sec. 4.2, We enhance the LRM by introducing ensemble-based technique to facilitate robust performance, uncertainty estimation and conservative update. Sec. 4.3 details the ReFL fine-tuning paradigm, with two key components: truncated mid-step setting and online alignment of LRM. The overall architecture of VELR is illustrated in Fig 2.

### 4.1 ARCHITECTURE AND TRAINING PARADIGM OF LATENT REWARD MODEL

**Network**. We first introduce the network architecture of LRM $\mathcal{R}_l(\boldsymbol{Z}^v, \boldsymbol{Z}^c) : \mathbb{R}^{C_l \times T_l \times H_l \times W_l} \times \mathbb{R}^{d_t \times d_c} \to \mathbb{R}$, with $\boldsymbol{Z}^v$ representing the latent variable $z_0$ in matrix form, and $\boldsymbol{Z}^c$ denotes the embedding of the prompt $c$. The LRM architecture combines 3D residual convolutions with a Transformer encoder to predict scalar rewards from latent representations. The design focuses on the local

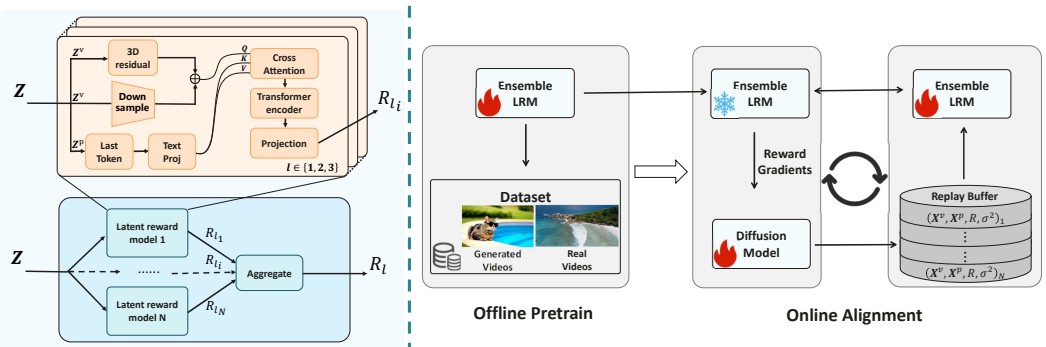

Figure 2: **The overall architecture of VELR.** Left: ensemble-based LRM architecture. Right: training procedure, the ensemble-based LRM is first pre-trained on a dataset and then used to guide diffusion model updates.

spatiotemporal structure with the capacity to capture long-range dependencies, providing a compact yest expressive architecture. Besides, the textual features are extracted from the prompt by the text encoder and the prompt embeddings are injected into the video features by cross-attention modules to ensure text-video alignment.

To be specific, given videos $\boldsymbol{X}^v \in \mathbb{R}^{C \times T \times H \times W}$ is fed to the VAE encoder of the diffusion model to obtain pure latent representations $\boldsymbol{Z}^v \in \mathbb{R}^{C_l \times T_l \times H_l \times W_l}$. The latent representations are first processed by a stack of 3D residual blocks to obtain the video features $f_v$. Each block applies two 3D convolutions with group normalization and residual connection:

$$\boldsymbol{H}^{(l)} = \sigma \left( GN \left( \boldsymbol{W}_2^{(l)} * \sigma \left( GN \left( \boldsymbol{W}_1^{(l)} * \boldsymbol{H}^{(l-1)} \right) \right) \right) + \boldsymbol{D}(h^{(l-1)}) \right) \qquad (4)$$

where $l \in \{1, 2, 3\}$ represents the list of blocks, $\boldsymbol{H}^{(0)} = \boldsymbol{Z}^v$, $\boldsymbol{W}_1^{(l)}$ and $\boldsymbol{W}_2^{(l)}$ represent the convolutional blocks, $*$ denotes 3D convolution, $\sigma$ is SiLU, and $\boldsymbol{D}$ is a downsampling mapping. The resulting feature $\boldsymbol{H}^{(3)}$ is reshaped into a sequence of video features $\boldsymbol{Z}^{vf} \in \mathbb{R}^{(T_l H_l W_l) \times d_v}$, with $d_v$ being the feature dimension. Then the video feature is augmented with temporal and spatial positional embeddings $\boldsymbol{Z}^{vf} = \boldsymbol{Z}^{vf} + \boldsymbol{P}_v^\phi$, with $\boldsymbol{P}_v^\phi$ being the learnable position embeddings.

We then employ cross attention to achieve cross-modal alignment between video features and textual prompts (Gorti et al., 2022). Specifically, cross attention allows video representations to selectively aggregate semantic information from the text while preserving their spatiotemporal structure, ensuring consistency with the prompt and avoiding the computational overhead of concatenating modalities. The prompt $p$ is first embedded to text embeddings $\boldsymbol{Z}^c \in \mathbb{R}^{d_t \times d_c}$, with $d_t$ being the number of tokens and $d_c$ being the dimension of prompt features. To keep the network lightweight, we follow CLIP (Radford et al., 2021) and use only the last token of the embedding as input $\boldsymbol{Z}_{d_t,:}^c \in \mathbb{R}^{1 \times d_c}$. This last token captures most of the information while significantly reducing the data dimensionality. The last text feature $\boldsymbol{Z}_{d_t,:}^p$ is projected to the video feature dimension $d_v$ to obtain the key and value in the cross attention module $\boldsymbol{P}_f = \boldsymbol{Z}_{d_t,:}^p \boldsymbol{W}_p$, with $\boldsymbol{W}_p \in \mathbb{R}^{d_p \times d_v}$. The cross attention block is represented as below:

$$\text{Attn}(\boldsymbol{Z}^{vf}, \boldsymbol{Z}_{d_t,:}^c, \boldsymbol{Z}_{d_t,:}^c) = \text{softmax}(\frac{\boldsymbol{Z}_{d_t,:}^c (\boldsymbol{Z}_{d_t,:}^c)^T}{\sqrt{d_v}}) \boldsymbol{Z}^{vf}. \qquad (5)$$

Then the outputs are passed to a multi-head self-attention and feedforward layers. Finally, we aggregate the token representations and predict the scalar reward $R_l^\phi$.

**Offline Training of LRM**. To train the latent reward model, we adopt the method of training the LRM on a dataset $\mathcal{D} = \{\boldsymbol{Z}_i^v, \boldsymbol{Z}_i^c\}_{i=1}^N$. We adopt the dataset of MVQA-68K (Pu et al., 2025) which is composed of real high-quality videos, as detailed in Appendix C.1. We also construct a dataset composed of generated videos based on the diffusion model. In addition, we generate a complementary dataset using samples produced by the diffusion model. The real videos provide rich

visual and semantic diversity, which enhances the generalization ability and robustness of the LRM. Meanwhile, the generated samples ensure that the LRM is well aligned with the latent distribution induced by the diffusion model, thereby enabling it to accurately evaluate outputs during video generation. The combination of these two sources of data allows the LRM to simultaneously capture the distributional characteristics of generated samples and retain reliable performance on real data.

We train the latent reward model based on this dataset with huber loss. Huber loss combines the benefits of mean square error (MSE) and mean absolute error (MAE): it behaves like MSE for small errors, enabling precise fitting, and like MAE for large errors, making it robust to outliers. Compared to MSE, it stabilizes training and improves convergence, especially on noisy data.

$$\mathcal{L}_{lrm}^{\delta}\left(\mathcal{R}(\text{Dec}(\cdot)), R_l^{\phi}(\cdot)\right) = \mathbb{E}_{\boldsymbol{Z}_i^v, \boldsymbol{Z}_i^p \in \mathcal{D}}\left[\begin{cases} \frac{1}{2}\left(\Delta(z)\right)^2, & \text{if } |\Delta(z)| \leq \delta \\ \delta\left(|\Delta(z)| - \frac{1}{2}\delta\right), & \text{if } |\Delta(z)| > \delta \end{cases}\right] \tag{6}$$

where $\text{Dec}(\cdot)$ represents the VAE decoder and $\Delta(z) = \mathcal{R}(Dec(\boldsymbol{Z}_i^v), \boldsymbol{Z}_i^p) - R_l^{\phi}(\boldsymbol{Z}_i^v, \boldsymbol{Z}_i^p)$.

## 4.2 ENSEMBLE-BASED LATENT REWARD MODEL

We first note that the current LRM exhibits unsatisfactory predictive performance: it struggles to accurately align with the ground-truth reward model, showing noticeable errors on both the training and test sets (as later confirmed in Sec. 5.6). This indicates that even within the in-distribution regime, the LRM provides only a coarse approximation of the true reward model.

This issue becomes even more pronounced under distribution shift. As the diffusion model inevitably drifts during fine-tuning, the input video latents—which are produced by the model's VAE encoder and DiT backbone—also shift accordingly. Consequently, the LRM frequently encounters samples that fall outside the offline distribution (OOD). These OOD samples include particularly important cases: videos that receive unusually high or low rewards, which are precisely the instances that guide ReFL updates toward desirable or undesirable directions. However, the LRM generalizes poorly on such OOD examples, exhibiting large deviations from the ground-truth reward.

To this end, we introduce ensemble technique of the LRMs. Ensemble techniques have been extensively studied in offline RL for estimating the Q-function, which can improve the model's ability, mitigate biases introduced by a single network, provide stronger robustness to out of distribution (OOD) data (An et al., 2021; Peer et al., 2021), which train several models separately and aggregate the outputs of these models to obtain the final output. For the LRMs, the ensemble technique is injected through the following equation:

$$\mathcal{R}_l(\boldsymbol{Z}^v, \boldsymbol{Z}^p) = \text{Agg}(\mathcal{R}_1(\boldsymbol{Z}^v, \boldsymbol{Z}^p), ..., \mathcal{R}_N(\boldsymbol{Z}^v, \boldsymbol{Z}^p)) \tag{7}$$

where $\mathcal{R}_i(\boldsymbol{Z}^v, \boldsymbol{Z}^p)(i \in 1, ..., N)$ represents the reward function trained independently, whereas Agg denotes an aggregation operator, which may be instantiated as the maximum, minimum, or mean operator. The ensemble techniques can enhance the expressive capacity of the learned LRM and improve robustness to OOD samples through collective predictions. Furthermore, the variance among ensemble members provides a natural estimate of prediction uncertainty: samples with high variance are more likely to lie outside the training distribution and therefore may require further alignment with the ground-truth reward. Moreover, given the relatively lightweight architecture of the LRM, employing ensemble techniques does not introduce significant computational overhead.

## 4.3 TRAINING PARADIGM OF VELR

After pretraining the ensemble-based LRM, we leverage it to fine-tune the diffusion model using the ReFL algorithm. For the loss function, we incorporate a KL divergence regularization term to mitigate reward hacking:

$$\mathcal{L}_{VELR} = -\mathbb{E}_{c \sim \mathcal{D}_c, z_0 \sim \pi_\theta(z_0|c)}\left[R_l(z_0, c) - \beta KL\left[\pi_\theta, \pi_{\theta_{old}}\right]\right] \tag{8}$$

where $\mathcal{D}_c$ represents the dataset of prompts and $\pi_{\theta_{old}}$ represents the initial model.

**Truncated Mid Step Setting**: we observe that updating the model only at the late, low-noise denoising steps has minimal effect on video generation, which makes the ReFL solution in VADER and ImageReward ineffective. This is because the sampling trajectory of the video generative model

quickly enters a low-noise regime where additional perturbations have negligible impact. For instance, in Wan-2.1-1.3B, adding noise after 30 denoising steps out of 50 produces almost no observable change (see Appendix E for results). Conversely, at early denoising steps, the video reward model exhibits limited discriminative ability: latents derived directly from the velocity field remain highly blurred and provide unreliable feedback. Taken together, these observations indicate that fine-tuning solely at the terminal low-noise stages or at the very early velocity-derived latents is ineffective. Motivated by this analysis, we propose a *truncated mid-step* setting: we randomly select a denoising step in the mid noise regime that has a significant impact on the video as the starting point for retaining gradients, then denoise for several subsequent steps until frames become relatively clear, after which velocity-field guidance is used to generate the final video. This strategy enables effective fine-tuning in mid-noise steps while mitigating the adverse effects of overly blurry generations. This paradigm also effectively reduces training time, as the number of denoising steps involved in training is significantly decreased. The training procedure is illustrated in Fig. 1.

**Online Alignment of LRM**. Empirically, we observe that without updating the LRM, the discrepancy between the latent and true rewards grows steadily throughout fine-tuning, indicating its poor generalization to OOD regions (Refer to Appendix C.6). This observation underscores the necessity of enabling the LRM to not only evaluate OOD samples but also maintain alignment with the ground-truth reward during ReFL training.

To this end, we maintain a replay buffer $\mathcal{B} = \{(X^v, X^p, R, \sigma^2)\}_N$, which stores the inputs required by the ensemble-based LRM, the corresponding pixel-space reward, and the variance of the LRM outputs. During diffusion model updates, new samples are inserted into the replay buffer. The buffer is organized according to the variance $\sigma^2$, and samples with higher variance replace those with lower variance, ensuring that more informative samples are retained. At fixed intervals during training, the LRM is further fine-tuned using the samples stored in the replay buffer. To prioritize informative data, we employ the variance magnitude $\sigma^2$ as the sampling weight, assigning higher probabilities to samples with greater uncertainty and thereby improving the accuracy of reward modeling. To control memory usage and maintain computational efficiency, we adopt a small-scale replay buffer, which our experiments demonstrate to be sufficient in practice.

## 5 RESULTS

In the experiments, we focus on the following aspects: (1) To what extent can VELR reduce memory consumption compared to standard ReFL algorithms? (2) Does VELR maintain performance comparable to standard ReFL despite the memory reduction? (3) Can VELR remain effective in scenarios where standard ReFL are computationally infeasible? (4) What are the contributions of the individual components of VELR to the overall effectiveness?

**Baselines.** We compare against the following methods:

- **Base models**. OpenSora 1.2 (Zheng et al., 2024) CogVideoX-1.5 (Hong et al., 2023; Yang et al., 2025) and Wan-2.1 (Wan et al., 2025) are current state-of-the-art open-sourced T2V diffusion models. We consider them as base models for ReFL and VELR.

- **ReFL algorithms**. VADER and ImageReward are state-of-the-art ReFL solutions. We adopt their updating paradigm and incorporate the LRM setting.

- **LRM-based ReFL**. Dollar (Ding et al., 2025) is the first ReFL solution based on LRMs. Though it is not open sourced and utilized on the consistency model, we believe it is an important baseline and we reproduce it based on its description.

**Reward models.** We use the following reward models to fine-tune the LDM.

- **CausalVQA** (Pu et al., 2025) is built on VideoAlign (Liu et al., 2025b) and fine-tuned on a diverse set of high-quality datasets, resulting in enhanced capabilities. We adopt it as the reward model baseline and use its datasets as data for the ensemble-based LRM.

- **UnifiedReward** (Wang et al., 2025b). We adopt UnifiedReward as one of our reward models and utilize its pointwise scoring functionality to provide fine-grained, human-aligned evaluation for video generation.

Table 1: Memory consumption (GB) of different modules for two reward models during gradient backpropagation, the memory reduction and the percentage of total memory consumption.

| Method | CausalVQA | | | | UnifiedReward | | | |
|---|---|---|---|---|---|---|---|---|
| | RM/LRM | Decoder | DiT | Total | RM/LRM | Decoder | DiT | Total |
| ReFL | 32.61 | 54.76 | 18.39 | 105.76 | 105.44 | 54.76 | 18.39 | 178.59 |
| VELR | 4.79 | / | 18.39 | 22.18 | 4.79 | / | 18.39 | 22.18 |
| Reduction | 27.82 | 54.76 | 0.00 | 83.58 (20.97%) | 100.65 | 54.76 | 0.00 | 156.41 (12.42%) |

**Original**  **Vader**  **VELR**

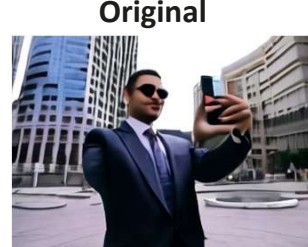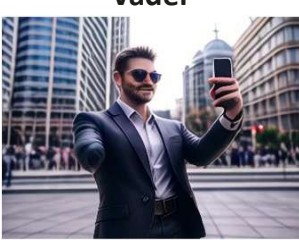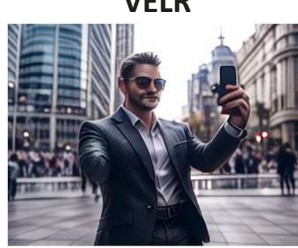

A man in a trendy suit taking a selfie in a city square, surrounded by modern buildings and a fountain.

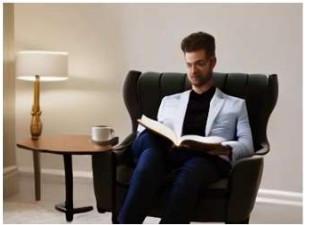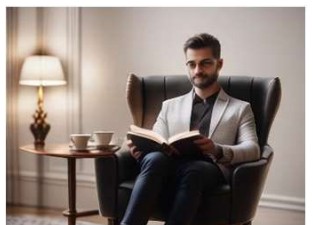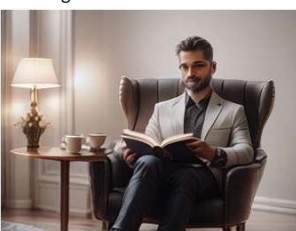

A man in a smart casual outfit sitting in a plush armchair, reading a book with a cup of coffee on a side table.

Figure 3: **Comparison of VELR and VADER on an Image RM PickScore.** The first, middle, and last columns correspond to the videos generated by the original model, VADER, and VELR, respectively. For fairness, all methods are trained for the same number of steps.

## 5.1 MEMORY OPTIMIZATION

To illustrate the substantial memory consumption of the standard ReFL algorithm and the memory efficiency of VELR framework, we conduct a detailed memory experiment. Under the Wan-2.1 setting, the batch size is set to 1. For ReFL, we apply multiple memory optimization techniques, as described in the Appendix C.3. Notably, the video RMs in ReFL processes only a single video frame at a time, whereas our VELR framework directly feeds the entire latent sequence into the LRMs without any compromise. Table 1 presents a detailed breakdown of memory usage across different modules, which demonstrates that VELR significantly reduces memory consumption while maintaining full-sequence modeling, highlighting its practical efficiency.

## 5.2 PERFORMANCE COMPARISON WITH STANDARD REFL

To assess how well the VELR paradigm preserves ReFL performance, we start with image RM experiments, as the standard ReFL algorithm is only applicable in this setting. We train the ReFL and VELR on OpenSora-1.2 using the PickScore RM (Kirstain et al., 2023). The memory consumption is reduced from 38.89GB to 19.73GB, achieving a **50% reduction**. Fig. 3 reports the fine-tuning results. The results demonstrate that, despite a substantial reduction in memory usage, VELR evolves in the same direction as standard ReFL, indicating that ensemble LRM effectively aligns with and leverages the original reward model. In addition, VELR produces richer details and higher video quality, suggesting that it progresses faster under the same number of training steps.

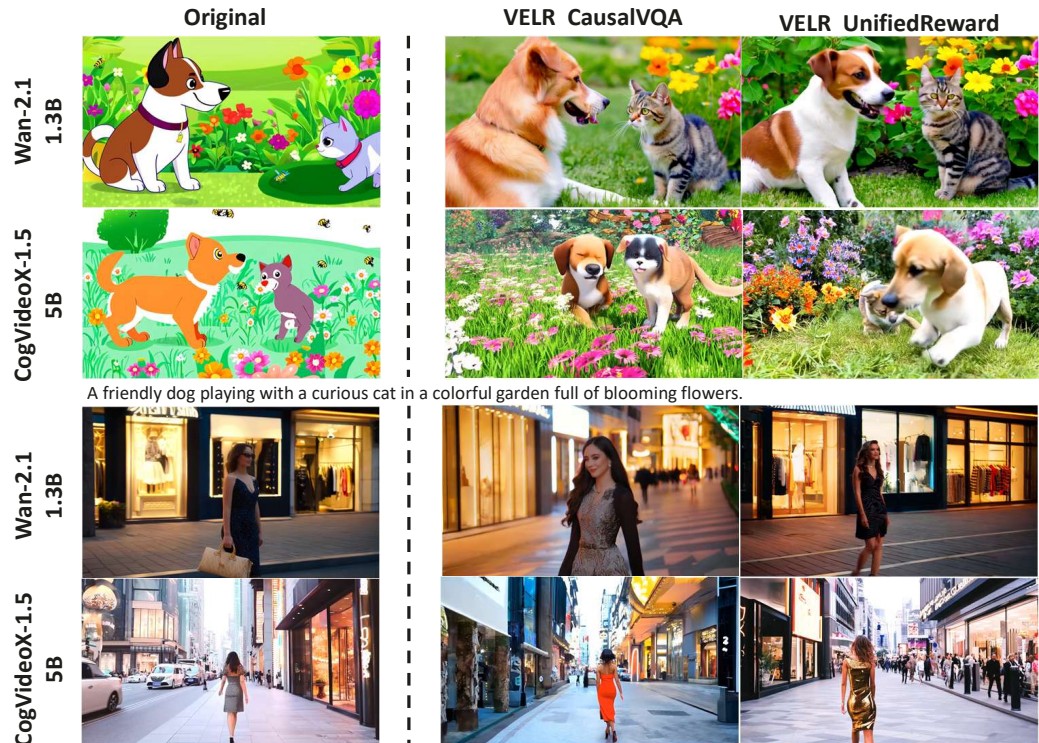

Figure 4: **Results of VELR on Wan-2.1 and CogVideoX-1.5.** Each pair of rows corresponds to the same prompt: the first row shows results from Wan-2.1, and the second row from CogVideoX-1.5.

### 5.3 VELR WITH VIDEO REWARD MODELS

Furthermore, we scale the VELR paradigm to scenarios where the standard ReFL algorithm is infeasible. This enables fine-tuning on more advanced and powerful diffusion models with larger and stronger reward models. To begin with, we provide a detailed comparison of the memory usage across different modules under various reward models (see Appendix C.3). The results show that, on average, the memory overload is reduced to **16.7%** of that required by the ReFL solution, significantly alleviating the memory burden. As shown in Fig. 4, on vRMs such as Wan-2.1 and CogVideoX-1.5, the standard ReFL algorithm fails to run, whereas VELR operates effectively and yields encouraging results. Compared to the base models, the fine-tuned outputs of VELR exhibit more realistic video quality and significantly stronger text–video alignment. For example, in the third prompt, the VELR vividly captures the notion of a busy street and generates various persons in the background, while the baseline Wan-2.1 model produces only a single person in the scene. Importantly, the LRM in VELR remains fixed and does not increase in size with larger reference reward models, underscoring the broad applicability of the VELR paradigm.

### 5.4 AUTOMATIC EVALUATION ON VBENCH

To evaluate the effectiveness of VELR on unseen metrics, we compare the fine-tuned and original models on the VBench (Huang et al., 2024) and VBench2.0 (Zheng et al., 2025) benchmarks. The results, summarized in Table 2, show that VELR consistently outperforms the original models, achieving an average improvement of 2.74, which demonstrates its superior generalization. Additional details regarding the evaluation protocols and metric computation are provided in Appendix C.5.

Table 2: Evaluation results across multiple metrics from both Vbench and VBench2.0.

| Methods | Overall Consistency | Aesthetic Quality | Human Fidelity | Composition | Image Quality | Average |
|---|---|---|---|---|---|---|
| Wan-2.1 1.3B | 22.89 | 64.01 | 81.76 | 38.15 | 66.41 | 54.64 |
| VELR Causal-VQA | 23.32 ↑ +0.43 | 64.74 ↑ 0.63 | 84.37 ↑ +2.61 | 41.79 ↑ +3.64 | 64.54 ↑ +0.53 | 55.55 ↑ +0.91 |
| VELR unified-reward | 23.35 ↑ +0.46 | 66.21 ↑ 2.20 | **86.85** ↑ +5.09 | 43.98 ↑ +5.83 | **69.62** ↑ +3.21 | 58.01 ↑ +3.37 |
| CogVideoX-1.5 5B | 27.71 | 62.53 | 61.92 | 43.85 | 65.26 | 52.25 |
| VELR Causal-VQA | 28.14 ↑ +0.43 | 61.94 ↓ -0.59 | 68.59 ↑ +6.67 | 46.29↑ +2.44 | 65.18 ↓ -0.06 | 54.03 ↑ +1.78 |
| VELR unified-reward | 31.26 ↑ +3.55 | 65.18 ↑ +2.65 | 71.04 ↑ +9.12 | 51.43 ↑ +12.28 | 66.87 ↑ +0.46 | 57.16 ↑ +4.91 |

## 5.5 HUMAN EVALUATION

To validate VELR's alignment with human preferences, we carried out a study to evaluate human preferences. The test consisted of a side-by-side comparison between VELR and original models. We include dimension about Video Fidelity, Semantic Consistency, Dynamic Degree and Aesthetic. Qualitative results, illustrated in Fig. 6, support that VELR consistently outperforms the original models and generates more text-aligned videos with higher video quality.

## 5.6 ABLATION STUDY

**Truncated mid-step setting**. We conducted extensive ablation studies on the two key hyperparameters of the truncated mid-step setting: the denoising step $N$ at which gradients start to be retained, and the truncated duration $K$. Each variant is denoted as VELR-N-K, and the results are shown in Fig. 5, which highlights the effectiveness of the truncated mid-step setting

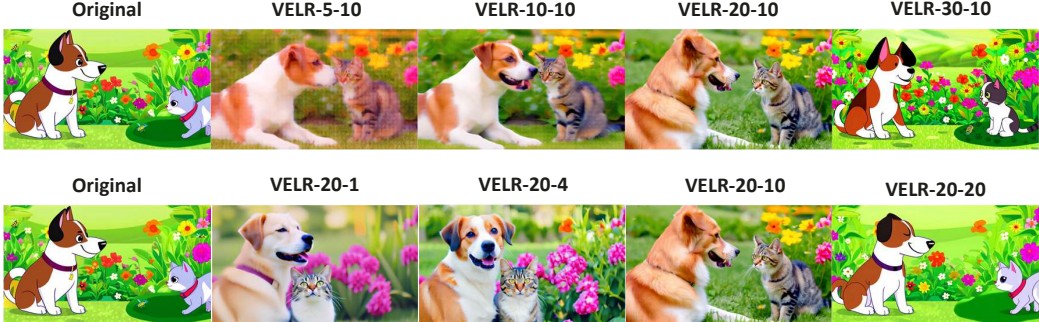

Figure 5: **Ablation study on the truncated mid-step setting.**

For the denoising start step $N$, we observe that choosing an early step leads to blurry outputs, as the T2V model is not yet capable of producing sharp frames—many details remain under-developed, and these blurry intermediate predictions ultimately guide the model toward blurry final videos. Conversely, setting $N$ too late results in minimal changes, consistent with the phenomenon discussed in Sec. 4.3: gradients from late steps have limited influence on the video content.

For the truncated length $K$, a small $K$ behaves similarly to an early $N$, producing blurry videos and unnatural lighting. A large $K$ accumulates gradients from many late denoising steps; however, late-step gradients overwhelmingly dominate but contain little meaningful information. As a result, the effective gradients from earlier informative steps are diluted, leading again to very limited updates to the T2V model.

Furthermore, we compared results within the 15–25 step range (see Appendix C.7) and found them largely similar, indicating that the truncated mid-step setting is insensitive to the choice of the effective interval—any step within this range yields good performance. In our experiments, we randomly select the denoising start step within this interval to promote better generalization. Moreover, the same hyperparameters are applied across all models and reward functions (see Appendix C.1), demonstrating the usability and robustness of the VELR paradigm.

**Effectiveness of ensemble-based LRM**. We conduct an ablation study on different LRM configurations, including the LRM proposed in Dollar (Ding et al., 2025), the non-ensemble variant described in Section 4.1, and ensemble-based LRM models with varying numbers of components (3, 5, and 10). All models are trained under identical settings, and their performance on both training and test sets is reported in Table 3. The results clearly demonstrate that ensemble-based LRM significantly outperforms the one described by Dollar. Furthermore, ensemble methods substantially enhance the representational capacity of latent reward models. The superior test performance also highlights that ensembles improve generalization and strengthen robustness against OOD samples.

For more ablation study on the LRM components and OOD generalization, see Appendix C.7.

Table 3: **The comparison of different LRM configurations.**

| LRM | Dollar | | LRM-en-1 | | LRM-en-3 | | LRM-en-5 | | LRM-en-10 (ours) | |
|---|---|---|---|---|---|---|---|---|---|---|
| | MSE | PCC | MSE | PCC | MSE | PCC | MSE | PCC | MSE | PCC |
| Train set | 0.832 | 0.59 | 0.315 | 0.79 | 0.159 | 0.85 | 0.072 | 0.92 | **0.003** | **0.98** |
| Test set | 1.215 | 0.34 | 0.851 | 0.68 | 0.490 | 0.77 | 0.158 | 0.84 | **0.008** | **0.91** |

**Training Paradigms**: We compare the impact of different ReFL update paradigms on Wan-2.1. Specifically, we consider the intermediate-step sampling paradigm proposed by ImageReward (Xu et al., 2023), the truncated-last-N-steps paradigm represented by Vader (Prabhude-sai et al., 2025). Because of the memory overload, we implement the LRM version of these paradigms. All above paradigms are trained with identical setting, only the ReFL update paradigm is different. We also include and the LRM-based baseline Dollar (Ding et al., 2025) The results are shown in Fig. 7. It can be observed that the first two approaches have limited influence on the intrinsic attributes of the generated videos. Dollar exhibits blurry background and disorganized subjects. In contrast, EVLR provides effective updates, with a noticeable improvement in video quality, fidelity, lighting and shadows.

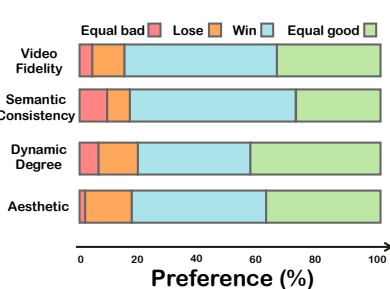

Figure 6: Human Evaluation

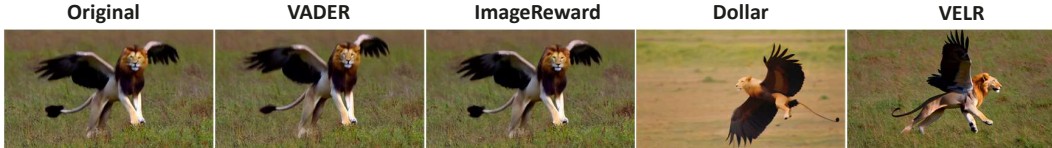

Original  VADER  ImageReward  Dollar  VELR

A lion with the wings of an eagle, soaring through the sky with majestic ease.

Figure 7: **Results of different ReFL update paradigm.**

## 6 CONCLUSION

In this paper, we introduced VELR, which leverages ensemble latent reward models to address the prohibitive memory demands of video reward model based ReFL algorithm. By enhancing the capacity of LRMs and incorporating ensemble estimation with conservative updates, our approach achieves efficient, scalable, and robust ReFL across large-scale T2V models. However, VELR still relies on truncated paradigms and does not fully exploit information from earlier denoising steps, which may limit optimization efficiency. In future work, we aim to develop ReFL algorithms that can stably leverage these earlier steps, potentially further improving training efficiency and enhancing overall model performance.

ETHICS STATEMENT

This work studies reward feedback learning for fine-tuning video generation models. To evaluate the quality of generated videos, we conducted human subjective evaluation. All participants involved in the evaluation were volunteers who were informed about the non-commercial and research-only nature of the study. No personally identifiable information was collected, and no potentially harmful or offensive content was generated. We believe our research complies with the ICLR Code of Ethics.

REPRODUCIBILITY STATEMENT

We have made significant efforts to ensure the reproducibility of our work. The proposed method is described in detail in Sec. 4 of the main paper, including model design and training procedure. The datasets used and their processing steps are documented in Appendix C.1, and the complete set of training hyperparameters is provided in Appendix C. Together, these materials provide sufficient information for reproducing our experiments. We commit to releasing the source code publicly after this round of submission.

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

## A  THE USE OF LARGE LANGUAGE MODELS (LLMs)

In the preparation of this paper, we used large language models (LLMs) as an auxiliary tool for language polishing, grammar checking, and improving the readability of sentences. No part of the research ideation or experimental design relied on LLMs. The authors take full responsibility for the final content.

## B  PSEUDO CODE

The pseudo code of the training pipeline of VELR is illustrated in Algorithm 1.

---

**Algorithm 1** Training Pipeline

---

**Require:** text-video offline dataset $\mathcal{D}_{off}$, prompt dataset $\mathcal{D}_p$, initial diffusion model parameter $\theta$, initial LRM parameter $\phi$, learning rate $\eta$, pre-train learning rate $\eta_{lrm}$, ODE solver $\Psi$, noise schedule $\alpha(t), \beta(t)$, VAE encoder Enc, decoder Dec, pixel-space RM $\mathcal{R}$, latent reward model $\mathcal{R}_l^\phi$, LRM training threshold $\delta$, tuncated step $K$, replay buffer $\mathcal{B}$.

  // Pre-train ensemble-based LRM
  **repeat**
    Sample $(\boldsymbol{X^v}, \boldsymbol{c}) \sim \mathcal{D}_{off}$
    $R = R(\boldsymbol{X^v}), R_l = R_l(\text{Enc}(\boldsymbol{X^v}), \boldsymbol{c})$
    compute $\mathcal{L}_{lrm}^\delta$ as Eqn. 6
    $\phi \leftarrow \phi - \eta_{lrm} \nabla_\phi \mathcal{L}_{lrm}^\delta$
  **until** convergence
  // Alternately update LRM $\mathcal{R}_l^\phi$ and diffusion model $\theta$
  $\theta^- \leftarrow \theta$
  **repeat**
    **if** update Diffusion Model **then**
      // update diffusion model with LRM
      Sample $\boldsymbol{c} \sim \mathcal{D}_p, n \sim \mathcal{N}[\boldsymbol{0}, \boldsymbol{I}]$
      **for** $i = T, T-1, \ldots, t_{mid}$ **do**
        Predict noise $\epsilon_\theta(z_i, i)$                       ▷ no gradient
        Compute denoised state $\hat{z}_i = f(z_i, \epsilon_\theta)$         ▷ no gradient
        Update latent with solver: $z_{i-1} \leftarrow \Psi(z_i, \hat{z}_i, i)$     ▷ no gradient
      **end for**
      $t_{end} \leftarrow t_{mid} - K$
      **for** $i = t_{mid} - 1, \ldots, t_{end}$ **do**
        Predict noise $\epsilon_\theta(z_i, i)$                       ▷ with gradient
        Compute denoised state $\hat{z}_i = f(z_i, \epsilon_\theta)$         ▷ with gradient
        Update latent: $z_{i-1} \leftarrow \Psi(z_i, \hat{z}_i, i)$         ▷ with gradient
      **end for**
      $\mathcal{L}_{KL} = KL[\epsilon_\theta(z_{t_{end}}, t_{end}), \epsilon_{\theta^-}(z_{t_{end}}, t_{end})]$   ▷ with gradient
      Predict the $z_0$ based on the velocity               ▷ with gradient
      compute $\mathcal{L}_{VELR}$ as Eqn. 8                   ▷ with gradient
      $\theta \leftarrow \theta - \eta \nabla_\theta \mathcal{L}_{VELR}$                   ▷ with gradient
      Store sample $(X^v, X^p, R, \sigma^2)$ into replay buffer $\mathcal{B}$
    **else**
      // align LRM with $\mathcal{R}$
      Sample $(X^v, X^p, R, \sigma^2) \sim \mathcal{B}$ with weight $\sigma^2$
      $R = R(\boldsymbol{X^v}), R_l = R_l(\text{Enc}(\boldsymbol{X^v}), \boldsymbol{c})$
      compute $\mathcal{L}_{lrm}^\delta$ as Eqn. 6
      $\phi \leftarrow \phi - \eta_{lrm} \nabla_\phi \mathcal{L}_{lrm}^\delta$
    **end if**
  **until** convergence

---

## C  Experiment and Hyperparameter Details

For all the qualitative or pairwise comparisons between different methods, we ensure to use the same random seed.

### C.1  Pre-training of ensemble-based LRM

As outlined in Sec. 4.1, our first step is to pretrain the ensemble-based LRM. We utilize bf16 precision and FlashAttention to improve efficiency.

**Dataset Construction.** For this stage, we adopt the MVQA-68K dataset (Pu et al., 2025), which contains a large collection of high-quality real-world videos. MVQA-68K is curated from several publicly available sources, notably Panda-70M (Chen et al., 2024b) and Koala-36M (Wang et al., 2025a), and covers a broad spectrum of scenarios such as human activities, urban landscapes, wildlife, vegetation, and indoor settings.

Since the videos we fine-tune on are of relatively high resolution (480p for Wan-2.1 and 720p for CogvideoX-1.5), we first filter out all videos in the dataset with a resolution lower than 480p. In addition, our experiments only consider landscape videos, as they generally yield better performance; thus, portrait videos are further removed. After applying these filtering rules, we obtain a final collection of 4.16K videos. It is worth noting that MVQA-68K dataset contains only raw videos without their corresponding prompts. To enable effective training of the latent reward model, and to maintain consistency with existing pixel-space reward models, we annotate this video set with textual prompts. Specifically, we employ the Qwen-VL-7B (Bai et al., 2025) model to generate the annotations, using the prompt template provided below.

Furthermore, to ensure that the latent reward model can accurately evaluate samples generated by diffusion models—i.e., to maintain alignment with pixel-space reward models on generated data—we augment the dataset with corresponding diffusion-generated samples. In total, we produce 500 generated videos generated by Wan-2.1 and CogvideoX-1.5. The prompts used for generation cover a diverse range of categories, including humans, animals, plants, as well as indoor and outdoor scenes. In these prompts, we also include the prompts from the ReFL fine-tuning prompt dataset. This is to ensure that the LRM can produce accurate reward during the initial stage of ReFl fine-tuning.

For the hyperparameters of training of the latent reward model, we train the LRM with batch size 4 (without gradient accumulation) and learning rate 2e-5 for Wan-2,1 and 1e-4 for CogVideoX-1.5. The training process lasts for about 3 epochs to converge, which takes about 18 hours.

### C.2  Alternately Training of LRM and Diffusion Model

For this stage, we adopt an alternating update strategy between the two models. Specifically, after every 10 epochs of diffusion model training, we perform two rounds of LRM alignment. During diffusion model updates, we use a batch size of 4, with a learning rate of 1e-5 for Wan-2.1 and 2e-5 for CogVideoX-1.5. We maintain a replay buffer of size 256 (our experiments show that this size is sufficient, as enlarging the buffer does not lead to further performance improvements). To update the buffer, we replace the samples with the smallest variance using those from the current batch, ensuring that the buffer always stores the most uncertain samples of the model. For the Wan-2.1 model, we set the mid step to be 20 and the truncated step to be 10; for the CogVideoX-1.5 model, we set the mid step to be 20 and the truncated step to be 10.

To mitigate reward hacking, we incorporate a KL divergence regularization term with a weight of 10. As for the LRM alignment step, it largely follows the same setup as the offline training phase, except that we adopt a smaller learning rate, uniformly set to 1e-5.

### C.3  More Detailed Study on the Memory Overlaod

To illustrate the advantages of the VELR on ReFL, we conduct a more detailed memory overload experiments. Under the Wan-2.1 setting, the batch size is restricted to 1. For the ReFL algorithm, we adopt all possible techniques to optimize memory usage. Specifically, we employ LoRA (Hu et al., 2022) fine-tuning with a rank of 256, use bf16 precision, and apply gradient checkpointing to the DiT module, the decoder, and the reward model. In addition, the reward model is offloaded to

the CPU when not in use, and the text embeddings of prompts are precomputed in advance, so the embedding module is not loaded during training.

In particular, to further reduce memory consumption, the video reward model in ReFL only takes a single video frame as input. This design means that part of the latent representation must first be decoded by the VAE decoder and then passed into the video reward model. By contrast, our proposed VELR framework feeds the entire latent sequence directly into the latent reward model, without any such compromise.

We provide a detailed breakdown of memory usage across different modules during backpropagation, as shown in Table 1. The results reveal that, despite the significant compromises and approximations adopted in ReFL—where the reward model only processes a single frame—the gradients of the reward model and the VAE decoder still dominate memory consumption. This results in high infrastructure demands for ReFL. In contrast, VELR effectively eliminates the gradients of these two modules while maintaining competitive performance, thereby demonstrating both its effectiveness and memory efficiency.

### C.4 COMPARISON WITH IMAGE REWARD MODEL

To further demonstrate the superiority of VELR with a video reward model, we compared the performance of VELR against ReFL using an image reward model. The results, shown in the figure, further highlight the effectiveness and significant contributions of our proposed paradigm.

### C.5 DETAILS OF VBENCH EVALUATION

Details of the dimensions we choose is shown below:

- **Overall Consistency**: Measures the alignment between generated videos and the provided textual prompts, assessing both semantic accuracy and stylistic coherence across the video sequence. Evaluation uses the ViCLIP model to compute similarity between video and text embeddings.
- **Aesthetic Quality**: Evaluates the perceived visual appeal of generated videos, considering color harmony, composition, and overall artistic quality. Frames are scored using a pretrained aesthetic predictor, then aggregated.
- **Human Fidelity**: Assesses the anatomical correctness and temporal consistency of human figures, ensuring realistic appearance and motion. Specialized models detect anomalies and evaluate identity consistency across frames.
- **Composition**: Measures the spatial arrangement and interaction of objects within the video. Evaluated via spatial relationship and multi-object arrangement metrics, ensuring proper placement and logical scene composition.
- **Image Quality**: Evaluates technical quality of individual frames, including sharpness, noise, and exposure. Assessed with an image quality predictor (e.g., MUSIQ) and aggregated for overall video quality.

We strictly follow the evaluation protocols of VBench and VBench2. Specifically, VBench generates five videos per prompt, while VBench2 generates three. All methods are evaluated under the same random seed to ensure a fair comparison.

### C.6 ERROR CURVES

To illustrate the consequences of not updating the LRM, we show the absolute error between the latent rewards and the true rewards during training, as depicted in the fig. 8

### C.7 MORE ABLATION STUDY ON LRM

**LRM Components**. To better illustrate the contribution of each component in the LRM, we conduct ablation studies by evaluating two variants: VELR/C (without the 3D CNN) and VELR/T (without the Transformer). Specifically, VELR/T removes the Transformer Encoder block and directly projects the output of the cross-attention module to the final reward predictions, whereas VELR/C

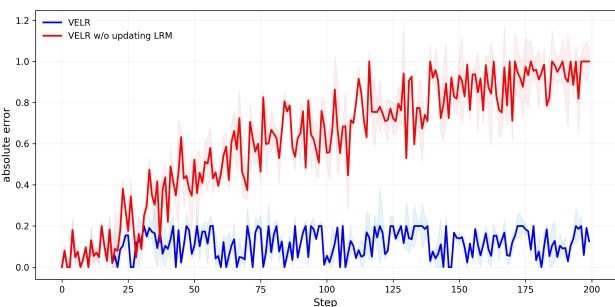

Figure 8: **The absolute error curve of the VELR**

removes the 3D CNN module and downsamples the video features directly to form the $Q$ and $K$ inputs for the cross-attention module. The estimation error and pearson correlation results are shown in table 4 and the generated videos are illustrated in Fig. 9

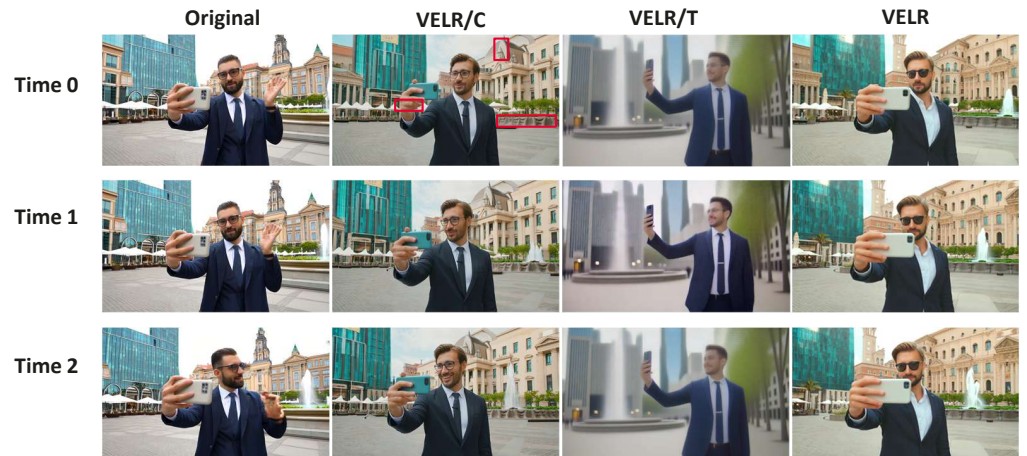

A man in a trendy suit taking a selfie in a city square, surrounded by modern buildings and a fountain.

Figure 9: **Video results of different LRM settings. Each row corresponds to the frame at the n-th second of the video, and each column represents a different variant.**

Table 4: **The comparison of different LRM settings on the train and test set.**

| LRM | VELR/T | | VELR/C | | VELR | |
|---|---|---|---|---|---|---|
| | MSE | PCC | MSE | PCC | MSE | PCC |
| Train set | 0.320 | 0.78 | 0.278 | 0.79 | 0.003 | 0.98 |
| Test set | 0.773 | 0.65 | 0.671 | 0.71 | 0.008 | 0.91 |

From the results, we can see that VELR performs substantially better than both variants in quantitative metrics: the variants yield larger MSE and smaller correlation coefficients, demonstrating the superior performance of the ensemble LRM proposed in VELR.

The fine-tuning results further reveal that removing the Transformer module has a severe impact on the LRM. After fine-tuning, VELR/T produces videos that are extremely blurry and exhibit strong flickering. VELR/C removes the 3D CNN module; although it can still improve video quality to some extent, many regions display abnormal details, as highlighted in the red boxes in Fig. 9.

In contrast, the videos fine-tuned with the ensemble LRM show clearly improved and more stable quality. This highlights the importance of both key components within the LRM.

**Long-term consistency of different LRM settings**. To directly assess whether these components in LRM help the LRM learn long-term consistency, we compare metrics related to consistency from VBench. The results are shown in Table 5.

Table 5: Evaluation results across long-term consistency metrics from VBench.

| Methods | Overall Consistency | Subject Consistency | Background Consistency |
|---------|---------------------|---------------------|------------------------|
| Wan-2.1 1.3B | 22.89 | 95.12 | 96.68 |
| VELR/T | 22.46 ↓ -0.43 | 94.56 ↓ -0.56 | 94.69 ↓ -1.99 |
| VELR/C | 22.92 ↑ +0.03 | 95.28 ↑ 0.16 | 96.80 ↑ +0.12 |
| VELR | 23.35 ↑ +0.56 | 95.96 ↑ +0.84 | 97.27 ↑ +0.59 |

Results show that VELR/T underperforms the base T2V model, while VELR/C yields only marginal gains. In contrast, VELR consistently achieves the best performance, demonstrating that the ensemble LRM architecture is essential for modeling long-term consistency.

**LRM Performance on an OOD dataset**. To better demonstrate the robustness of the ensemble LRM architecture against OOD samples, we evaluate the performance of different variants of LRM on an OOD dataset. Specifically, we choose the test set of Panda-70M (Chen et al., 2024b) as the OOD dataset, which contains 6,000 samples. The MSE error and Pearson correlation of different LRM versions on this dataset are reported in Table 6.

As shown, the ensemble LRM retains good predictive ability on OOD samples, while other variants degrade substantially in both correlation and error metrics. This confirms the effectiveness of the ensemble design for improving OOD generalization.

Table 6: **The comparison of different LRM configurations on an OOD dataset.**

| LRM | Dollar | | LRM-en-1 | | LRM-en-5 | | LRM-T | | LRM-C | | LRM-VELR | |
|-----|--------|------|----------|------|----------|------|-------|------|-------|------|----------|------|
| | MSE | PCC | MSE | PCC | MSE | PCC | MSE | PCC | MSE | PCC | MSE | PCC |
| OOD Test set | 1.478 | 0.31 | 0.964 | 0.42 | 0.490 | 0.49 | 0.410 | 0.57 | 0.387 | 0.62 | **0.249** | **0.72** |

## C.8 SENSITIVITY ANALYSIS OF TRUNCATED MID STEP SETTING

We conducted a sensitivity analysis of the truncated mid step setting and found that selecting the denoising step within the range of 15–25 yields similar results, demonstrating that the truncated mid-step setting is robust to the choice of step number, as shown in Fig. 10.

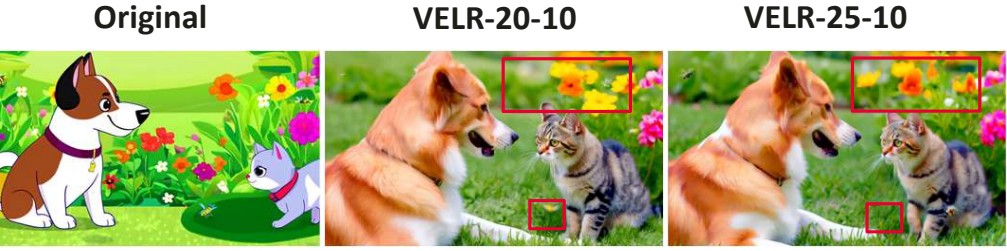

Figure 10: **Sensitivity results of VELR.**

The results show that the choice of intermediate steps has only a minor impact on the final outputs. Aside from subtle differences—such as the shape of flowers in the background or fine-grained details like the bee in the foreground or the dog's fur—the other regions remain highly similar. This demonstrates the robustness of VELR to the selection of intermediate steps.

## C.9 MODEL SIZES OF REWARD MODELS

The model scales of all reward models used in our work, including PickScore, Causal VQA, and Unified Reward, as well as the latent reward models, are summarized in Table 7.

Table 7: **The comparison of different LRM configurations on an OOD dataset.**

|  | PickScore | CausalVQA | UnifiedReward | EVLR(single) | EVLR(ensemble) |
|---|---|---|---|---|---|
| Parameters Size | 1B | 7B | 32B | 0.02B | 0.21B |

## D EXPERIMENT RESULTS

We list more results in the validation set as illustrated in Fig. 11

As shown in the figure, VELR significantly improves the plausibility of the videos and their alignment with the text. To be specific:

- For the first prompt, the concept of 'Splash of turquoise water' is more clearly expressed: compared to the nearly static pool in the original video, VELR produces a slow yet realistic splash motion that better aligns with the prompt.

- For the second prompt, the elephant becomes more prominent, the background is richer and more realistic, and the concept of 'bright sunlight' is highlighted, with both the elephant and the background exhibiting warm tones of sunlight.

- For the third prompt, the overall video style shifts from anime to realistic, with the background more effectively reflecting the idea of a 'fantasy landscape'.

- For the fourth prompt, the video style aligns more closely with the 'Vincent van Gogh' style described in the prompt, and the boat moves consistently forward, while in the original video the boat moves backward in the final second.

## E TEST-TIME SCALING EXPERIMENT

We conducted a Test-Time Scaling experiment. Specifically, we added random noise to the latent representations at different denoising steps (step = 0 corresponds to pure noise, and step = 50 corresponds to the real video). This operation allows direct exploration of the generation range of videos at different steps, as adding noise is a more direct means than fine-tuning. Following the setting of DanceGRPO, we applied noise according to the DDPM noise scheduler. The results of adding noise at different steps are shown in the fig. 12.

The results show that early steps significantly affect the entire video, while mid-stage noise mainly influences the subject and fine details. After step 20, the video remains largely unchanged.

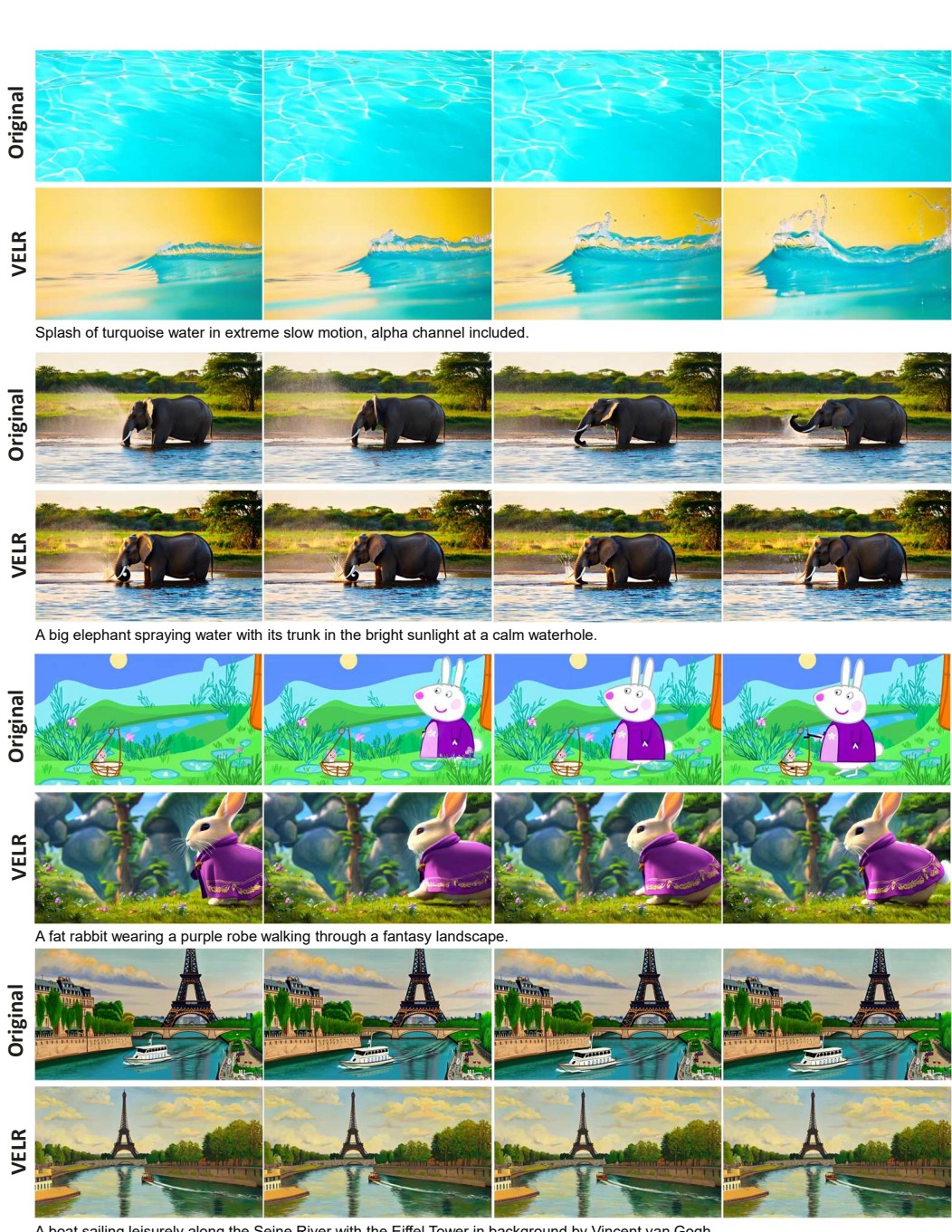

Splash of turquoise water in extreme slow motion, alpha channel included.

A big elephant spraying water with its trunk in the bright sunlight at a calm waterhole.

A fat rabbit wearing a purple robe walking through a fantasy landscape.

A boat sailing leisurely along the Seine River with the Eiffel Tower in background by Vincent van Gogh

Figure 11: **Results of VELR on the validation set.** Each pair of rows corresponds to the same prompt: the first row shows results from the original model, and the second row from VELR. Each row contains five frames from a video, evenly spaced in time.

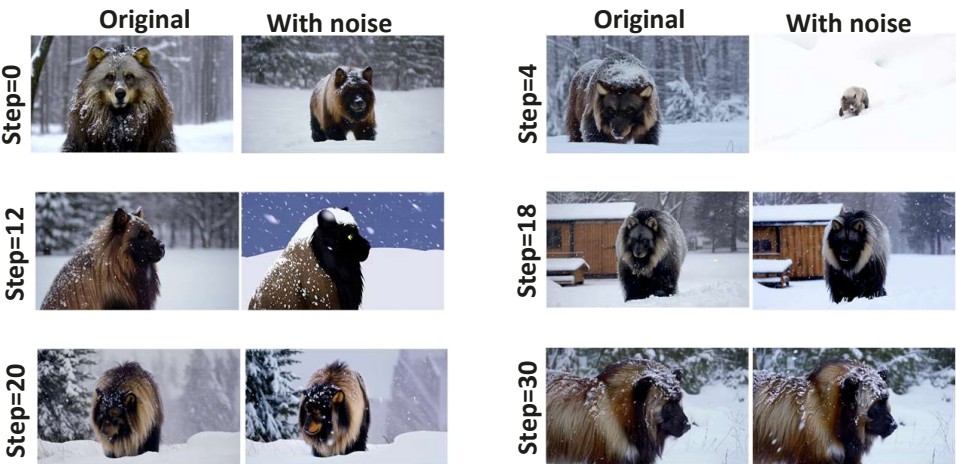

Figure 12: **Results of test-time-scaling.** Each row shows two steps: the left video is the original frame, and the right video is the noised frame.

