# OpenReview forum: "VELR: Efficient Video Reward Feedback via Ensemble Latent Reward Models"
_ICLR.cc/2026/Conference — ICLR 2026 Conference Withdrawn Submission_

### Official Review · Reviewer_u2cT · 2025-10-28

**Soundness:** 3
**Presentation:** 3
**Contribution:** 3
**Rating:** 4
**Confidence:** 4

**Summary:**

This paper proposes VELR (Efficient Video Reward Feedback via Ensemble Latent Reward Models), an innovative framework to overcome the prohibitive memory cost of applying large-scale video Reward Models (RMs) in Reward Feedback Learning (ReFL) for Text-to-Video (T2V) generation. By training an Ensemble Latent Reward Model (LRM) to predict rewards directly in the latent space, the framework successfully bypasses expensive backpropagation through the VAE decoder and the large video RM. The method achieves a substantial memory reduction (up to 150GB ) while maintaining comparable performance to standard ReFL, making previously infeasible Video RM-based fine-tuning a reality for large T2V models. The work is highly relevant and addresses a critical practical limitation in aligning video diffusion models.

**Strengths:**

1.	The proposed reward model enables the effective use of powerful, temporal-aware Video RMs (like UnifiedReward and CausalVQA) that were previously computationally inaccessible.
2.	The introduction of the Ensemble LRM is technically sound.
3.	Experimental results indicate the effectiveness of the proposed method.

**Weaknesses:**

1.	The primary benefit of moving from Image RMs to Video RMs is the improvement in temporal coherence. The paper critically lacks supplementary video material for the generated samples. Without visual evidence, it is impossible for the reviewer to verify the claimed improvements in temporal consistency and to judge the subjective quality, flicker, and artifacts of the generated videos. This omission severely undermines the empirical claims of the paper.
2.	While the paper compares against LRM-adapted baselines, there is no direct comparison against a dedicated, high-fidelity Image Reward model (running in its original pixel space). Additionally, Since Video RMs primarily prioritize temporal objectives, a convincing comparison is necessary to demonstrate that the VELR framework, despite its efficiency and temporal gains, fully retains the high perceptual visual quality boost provided by state-of-the-art Image RMs. The current results focus heavily on memory and speed, but not enough on the visual quality trade-off (if any) compared to the best image-focused alignment methods.
3.	While motivated, "Truncated Mid Step Setting" introduces a model-specific heuristic for selecting the "mid-step regime" that is dependent on the velocity prediction and noise schedule of the base T2V model (e.g., Wan-2.1 ). This reliance limits the general applicability and plug-and-play nature of the ReFL solution across different diffusion architectures.

**Questions:**

Please refer to Weaknesses for more details. If the concerns are solved, I will be glad to raise my score rate.

---

> ### Author Response · Authors · 2025-11-22
>
> We thank the reviewer for the constructive feedback and we are glad that the reviewer recognizes the soundness of the ensemble latent reward model, its ability to utilize temporal-aware video RMs, and the overall effectiveness of our approach. We will address the detailed comments point by point below.
>
> ## **Weakness 1:  Supplementary Video Material**
>
> Thank you for your suggestion. We have provided comparison videos on the anonymous project website (https://elvr-anonymous.github.io/VELR_anonymous/) and will continue to update them. We kindly refer the reviewer to the website for more details.
>
> ## **Weakness 2: Comparison against Image Reward Model**
>
> We respectfully clarify that we have **already conducted comparisons with a high-quality image RM** in the initial manuscript. As shown in Fig. 3 of Sec. 5.2, based on OpenSora 1.2, we perform two types of fine-tuning: (i) directly fine-tuning with the pixel-space image reward model PickScore2 (VADER), and (ii) estimating PickScore via the VELR framework and then fine-tuning. The results demonstrate that VELR achieves faster convergence while maintaining high visual quality.
>
> ## **Weakness 3: Truncated mid step setting introduces heuristic and limits plug-and-play of ReFL**
>
> - **Plug-and-play and heuristic of ReFL**: We respectively clarify that the previous ReFL methods are **NOT naturally plug-and-play**, as each adopts its own heuristic for step selection. For example, VADER [1] truncates the last ten steps and ImageReward [2] randomly samples from the last few steps.
>
> - **Advantage of truncated mid step setting**: while our truncated mid step setting introduces heuristic, it performs substantially better than previous ReFL approaches on sota T2V models (shown in Tab. 2 of Sec. 5.4 and Fig. 6 of Sec. 5.5) and can rapidly and effectively improve video quality (shown in Fig. 3 of Sec. 5.2).
>
> - **Universality of truncated mid step setting**: we apply the **SAME** hyperparameters across three T2V models and three reward models, and VELR performs consistently well. Thus, VELR’s general applicability is comparable to that of prior ReFL variants.
>
> - **Robustness to hyperparameter settings**: we conducted an ablation study and found that selecting the denoising step within the range of 15–25 yields similar effectiveness, demonstrating that the truncated mid-step setting has a good tolerance (see Appendix C.8).
>
>
> **References**
>
> [1] Prabhudesai et al. VADER: Video diffusion alignment via reward gradients. Arxiv 2025.
>
> [2] Xu et al. Imagereward: Learning and evaluating human preferences for text-to-image generation. NeurIPS 2023.

---

### Official Review · Reviewer_QQue · 2025-11-01

**Soundness:** 3
**Presentation:** 3
**Contribution:** 3
**Rating:** 6
**Confidence:** 3

**Summary:**

This paper addresses the mismatch between image-based reward models and the temporal objectives of text-to-video generation in Reward Feedback Learning (ReFL). The authors propose VELR, a memory-efficient framework that replaces direct video reward modeling with ensemble latent reward models operating in the latent space. By predicting rewards without backpropagation through VAE decoders and full video RMs, VELR substantially reduces memory usage while maintaining comparable performance. The ensemble design is intended to increase capacity, quantify uncertainty, and mitigate reward hacking. Experiments on OpenSora, CogVideoX-1.5, and Wan-2.1 with large-scale video RMs support the claim that VELR enables robust, scalable, and efficient ReFL for text-to-video at previously unattainable scales.

**Strengths:**

1. The manuscript is clearly written and accessible.
2. The motivation is well articulated; the work explores an effective video latent reward model.
3. The topic is important: latent reward models can reduce RL training costs, particularly for video generation.

**Weaknesses:**

1. Please report model sizes: the parameter count of the reward model, and the parameter count for the ensemble-based LRM (per model and overall).
2. Additional methodological details would improve clarity. In Section 4.3, when fine-tuning the LRM using outputs from the diffusion model:
    - Is the dataset the same as that used for LRM pretraining? If only diffusion-generated videos (no real videos) are used, how do you ensure training reliability and avoid distribution shift?
    - For the Truncated Mid Step setting, which intermediate steps are selected and why?
    - For the ensemble of LRMs, are all models trained on the same dataset and loss? If so, how is diversity encouraged to yield distinct rewards for each LRM in the ensemble LRMs?
3. Did you initialize any components from CausalVQA or UnifiedReward pretrained weights?
4. In Figure 3, the proposed method appears to offer limited improvement over the VADER baseline.
5. It would be better to include qualitative comparisons against the baseline Dollar.
6. Please clarify what dimensions the LRM's reward captures: is it decomposed into image quality, temporal consistency, semantic alignment, etc., or is it a single undifferentiated score?

**Questions:**

Typo on line 77: "ignificantly" should be "significantly".

---

> ### Author Response · Authors · 2025-11-22
>
> We thank the reviewer for the thoughtful and encouraging feedback. We are glad that the reviewer finds our manuscript clear and accessible, appreciates the motivation, and recognizes the benefits of our memory-efficient latent reward design and ensemble approach. We address the detailed concerns point by point below.
>
> ## **Weakness 1: Model sizes of Video RMs and Latent RMs**
> Thank you for the detailed question. The model scales of all reward models used in our work are summarized in the table below, including PickScore, Causal VQA, and Unified Reward, as well as the latent reward models. This table has been added to the revised manuscript to improve clarity and completeness.
> |Model| PickScore | CausalVQA | UnifiedReward| VELR(single)| VELR(ensemble)
> |-|-|-|-|-|-|
> | Parameters|1B|7B|32B|0.02B|0.21B
>
> ## **Weakness 2.1: Methodological details: dataset used for training LRM and methods to ensure training reliability**
> **Are the datasets used for LRM pre-training and fine-tuning the same?**
>
> No. The datasets used in these two stages are different. Specifically, LRM is **pre-trained** on a mixed dataset that contains both real videos and diffusion-generated videos. During ReFL training, however, LRM is **fine-tuned** on a dynamic replay buffer that consists exclusively of videos generated by the evolving diffusion model.
>
> **How to ensure training reliability and avoid distribution shift?**
>
> The above pre-training & fine-tuning design choice was made exactly to **ensure the training reliability and avoid the distribution shift** where the online **fine-tuning** stage helps the **pre-trained** LRM to better align with each generative model’s own generation process. Inspired by your question, we have polished the manuscript by highlighting the “Online Alignment of LRM” part in Sec. 4.3. Please kindly refer to it for more details.
>
> ## **Weakness 2.2: Methodological details: intermediate steps selection**
> We use an intermediate step region from 15 to 25 so that effectively affects video quality while avoids the high-noise region, which is difficult to control, or the low-noise region, which has limited impact on the video.
>
> Inspired by your question, we have provided further ablation studies in Sec. 6.4 to validate the truncated mid step setting. The results indicate that using too early denoising steps leads to blurry videos and using too late steps results in minimal video changes. Moreover, selecting the denoising step within the range of 15–25 yields similar results (see Appendix C.8). These results indicate that mid step selection is reasonable and robust.
>
> ## **Weakness 2.3: Methodological details: are all ensemble LRMs trained on the same dataset and loss?**
> Yes, all ensemble LRMs are trained on the same dataset and with the same loss.
>
> To encourage diversity, we initialize each LRM with a different random seed, which is a common practice in ensemble learning [1-2]. Our experiments show that this lightweight strategy is sufficient for producing distinguishable reward predictions across LRMs.
>
> ## **Weakness 3: Initialize any components from weights of the pretrained vRM?**
> No, we did not initialize any components of LRM from weights of vRM. The architecture of our LRM does not align with the pretrained vRM architectures and is initialized randomly.
>
> Despite this, the LRM is lightweight and trains efficiently—on 8 NVIDIA A100-80GB GPUs, training completes in approximately 2 hours of wall-clock time.
>
> ## **Weakness 4: Similar results against the baseline VADER?**
>
> Yes. We would like to highlight that the key motivation of our work is to **reduce memory usage and improve training speed of ReFL while maintaining effectiveness compared to original reward models**. The purpose of Fig. 3 is to show that VELR (our work) preserves similar quality compared to the original reward model. This observation precisely supports our contribution.
>
> ## **Weakness 5: Qualitative comparison against the baseline Dollar**
>
> Thank you for your insightful suggestion. We have added qualitative video comparisons with Dollar in Fig. 7. The results show that videos generated by Dollar exhibit disorganized subjects and blurry details, whereas videos generated by VELR have clear subjects, rich details, and realistic, plausible lighting and shadows.
>
>
> ## **Weakness 6: Dimension of the LRM’s reward**
>
> We use a **single, undifferentiated score** as the reward signal. We thank the reviewer for the suggestion and have added a detailed description of the reward signal in the manuscript.
>
> ## **Question 1: Typo of the content**
>
> We thank the reviewer for the careful reading. The typo on line 77 (“ignificantly”) has been corrected to “significantly,” and we have also checked and corrected other grammatical errors throughout the manuscript.
>
> **References**
>
> [1] B Lakshminarayanan et al. Simple and Scalable Predictive Uncertainty Estimation using Deep Ensembles. Neurips 2017.
>
> [2] S Fort et al. Deep Ensembles: A Loss Landscape Perspective. Arxiv 2020.

---

### Official Review · Reviewer_geYU · 2025-11-01

**Soundness:** 2
**Presentation:** 3
**Contribution:** 2
**Rating:** 4
**Confidence:** 3

**Summary:**

This paper addresses this problem: the prohibitive memory and computational cost of ReFL when using large-scale video reward models. The authors propose VELR, an efficient framework centered around an ensemble of Latent Reward Models. The core idea is to bypass the costly backpropagation through the VAE decoder and the pixel-space vRM by training LRMs to predict rewards directly from the diffusion model's latent representations. The paper claims that this approach, enhanced by ensemble techniques for robustness and a "truncated mid-step" training strategy for efficiency, significantly reduces memory consumption (by up to 150GB) while achieving performance comparable to standard ReFL methods.

**Strengths:**

- Quantifiable achievements like reducing memory usage to as low as 12.4% of the standard ReFL baseline demonstrate the framework's efficiency.
- While individual components (LRM, Ensemble, Replay Buffer) are not novel in isolation, their combination into a cohesive system to solve this specific problem works well.

**Weaknesses:**

1. Limited Methodological Novelty: The concept of LRM was previously introduced, and the use of ensemble learning to improve robustness and OOD generalization is a standard, widely-used technique in machine learning. The novelty is confined to the specific application and combination of these existing ideas.
2. Insufficient Analysis of "Proxy Risk" and Reward Distortion: The LRM is fundamentally an imperfect proxy for the true vRM, which introduces a critical risk: the optimization may be guided by the LRM's prediction errors. The paper identifies "reward hacking" but fails to deeply analyze this "proxy risk". The entire framework's stability rests on the fidelity of the LRM, yet there is no quantitative analysis of the error between the LRM's predictions and the vRM's true scores, nor a discussion of how this error might be amplified during training.
3. Methodological Choices Lack Rigorous Justification: Several key design decisions appear to be based on heuristics rather than thorough empirical or theoretical backing.
     - The "Truncated Mid-Step Setting" is justified by intuitive observations ("early steps are blurry, late steps have minimal effect") but lacks a proper ablation study. The paper does not provide experiments comparing different start points (t_mid) or backpropagation durations (K) to prove that their chosen "mid-step" range is truly optimal or generalizable.
     - The LRM architecture (3D CNN + Transformer) is asserted without ablative evidence. It is unclear if both components are necessary or if simpler architectures could suffice.

**Questions:**

1. **Proxy Risk**: Could you provide a quantitative analysis of the "proxy error" (e.g., correlation, MSE, or error distribution) between the ensemble LRM and the ground-truth vRM on an out-of-distribution test set? How does your framework ensure that this inherent error is not amplified during fine-tuning, potentially leading the diffusion model towards undesirable optima that only the LRM, not the vRM, finds rewarding?
2. **Truncated Mid-Step Setting**: To validate this strategy, could you provide an ablation study on the choice of the starting step t_mid and the number of backpropagation steps K? How sensitive is the final performance and training efficiency to these hyperparameters, and how does this justify that the "mid-step" is a fundamentally better strategy than, for example, a truncated early- or late-stage update?
3. **Spatio-Temporal Control**: While the LRM architecture is designed to capture spatio-temporal features, can you provide more direct evidence or analysis showing that it is indeed effectively guiding the temporal consistency of the generated videos? For instance, does an LRM trained with this architecture outperform one without the Transformer component in predicting rewards related to long-term consistency?

---

> ### Author Response · Authors · 2025-11-22
>
> We thank the reviewer for the constructive feedback and we are encouraged that the reviewer highlights the practical efficiency of VELR, including the substantial memory savings and the effective combination of latent reward models, ensemble methods, and replay strategies. We respond to the reviewer’s detailed comments in two parts below.
>
> # Part 1
>
> ## **Weakness 1: Methodological Novelty**
>
> Though the individual components have been studied previously, the novelty of VELR lies in integrating them into the first scalable framework tailored for large RMs. This framework directly addresses the memory bottlenecks that make ReFL impractical.
>
> Specifically:
>
> - We introduce an ensemble LRM that increases modeling capacity, provides uncertainty estimates, and mitigates reward hacking, thereby improving robustness for video reward modeling. To our knowledge, ensemble LRMs have not been thoroughly explored before.
>
> - We design a training procedure tailored for the ensemble LRM ReFL framework with two key designs: truncated mid step setting and online alignment of LRM, which has not been studied before.
>
> - Combined, VELR achieves up to **87.6% memory reduction** (please see Tab. 1, Sec. 5.1) so that unlocks the application of video RMs that were previously computationally infeasible. VELR receives a **remarkable gain in preference rate (41% vs. 8%)** in the human evaluation (please see Fig.6, Sec. 5.5). This demonstrates that the combination of these ideas has a practical impact beyond isolated methods.
>
> We have clarified these contributions in the revised manuscript in Sec. 1 to better highlight the novelty and significance of our work.
>
> ## **Weakness 2 & Question 1: Quantitative Analysis of "Proxy Risk"**
>
> **Proxy Error of LRM on train and test set**
>
> We would like to clarify that we do pay careful attention to mitigating the adverse effects of distribution shift and our **original submission** has already taken measures to stabilize the training process via an online alignment of LRM (with quantitative analysis provided in Fig. 8, Appendix C.6). Inspired by the suggestion, we have polished the manuscript to highlight this “Online Alignment of LRM” part in Sec. 4.3. Specifically:
>
> - **Online alignment of LRM**: VELR continuously aligns the LRM with the vRM during ReFL: samples with high LRM uncertainty are added to a replay buffer and retrained throughout ReFL training. This reduces distribution shift and helps correct prediction errors.
>
> - **Fidelity of the LRM through ReFL training**: a quantitative analysis of LRM prediction error throughout ReFL training was provided in Fig. 8, Appendix C.6. The results show that the estimation error remains consistent as training progresses, indicating that VELR effectively prevents proxy error.
>
> **Proxy Error of LRM on an OOD test set**
>
> We greatly appreciate the reviewers’ suggestions. We have added experiments evaluating the LRM’s proxy error on an OOD test set. Specifically, we choose  the test set of Panda-70M [1] as the OOD dataset, which contains 6,000 samples. We reported MSE and Pearson correlation (PCC) between the ensemble LRM and vRM. The results are summarized in the table below.
>
> |Model| Dollar | VELR-en-1 | VELR-en-5 | VELR/T | VELR/C | VELR-en-10 (ours)
> |:-:|:-:|:-:|:-:|:-:|:-:|:-:|
> | MSE $\downarrow$|1.48|0.97|0.63|0.41|0.39|**0.25**
> | PCC $\uparrow$|0.31|0.42|0.49|0.57|0.62|**0.72**
>
> As shown, the ensemble LRM retains good predictive ability on OOD samples, while other variants degrade substantially in both correlation and error metrics. This confirms the effectiveness of the ensemble design for improving OOD generalization.

---

> ### Author Response · Authors · 2025-11-22
>
> # Part 2
>
> ## **Weakness 3 & Question 2 & Question 3: Methodological Choices**
>
> **Ablation Study on "Truncated Mid-Step Setting"**
>
> Thank you for your valuable comments. We have provided more qualitative results in Sec. 5.6 which illustrate that the truncated mid-step setting is reasonable, robust, and generalizable.
>
> - **Reasonable**: The results indicate that using too early denoising steps leads to blurry videos, and using too late steps results in minimal video changes. Moreover, using too small truncated duration leads to blurry background and abnormally colored outline. Using too large truncated duration leads to limited video changes, which is because the gradients from late steps dominate all gradients.
>
> - **Robust**: Selecting the denoising step within the range of 15–25 yields similar effectiveness, demonstrating that the truncated mid-step setting provides good tolerance for the choice of steps.
>
> - **Generalizable**: We apply the same hyperparameters across three generative models and three reward models, consistently achieving strong performance, which further supports the generality of our approach.
>
> **Ablation Study on LRM components**
>
> Thank you for your valuable comment. We have added further ablation studies to evaluate the necessity of both components (the 3D CNN and the Transformer) in the LRM architecture. We introduce two variants: VELR/C (without 3D CNN) and VELR/T (without the Transformer). The estimation error and Pearson correlation are shown below:
>
> | Model | VELR/T | | VELR/C | | VELR ||
> |:-:|:-:|:-:|:-:|:-:|:-:|:-:|
> | metric | Train set | Test set | Train set | Test set | Train set | Test set |
> | MSE $\downarrow$ | 0.320 | 0.773 | 0.278 | 0.671 | 0.003 | 0.008 |
> | PCC $\uparrow$ | 0.78 | 0.65 | 0.79 | 0.71 | 0.98 | 0.91 |
>
> We also train the T2V model based on these variants. More qualitative results and details are provided in Appendix C.7.
>
> In conclusion, both variants show **clear drops** in reward prediction and Pearson correlation, and the generated videos show low visual quality, blurry details, abnormal lighting and shadow, indicating that each component contributes critically to capturing spatio-temporal structure.
>
> **Spatio-Temporal Control: Long-term consistency of different LRM settings**
>
> Following your suggestion, we compare metrics related to consistency from VBench to directly assess whether these components of LRM help learn long-term consistency.
>
> |Model| Wan 2.1 | Wan 2.1+VELR/T | Wan 2.1+VELR/C | Wan 2.1+VELR
> |-|:-:|:-:|:-:|:-:|
> | subject consistency $\uparrow$ |95.12|94.56 (-0.43)|95.28 (+0.16)|**95.96 (+0.84)**
> | background consistency $\uparrow$ |96.68|94.69 (-1.99)|96.80 (+0.12)|**97.27 (+0.59)**
> |overall consistency $\uparrow$|22.89|22.46 (-0.43)|22.92 (+0.03)|**23.35 (+0.46)**
>
> Results show that VELR/T underperforms the base T2V model, while VELR/C shows only marginal gains. In contrast, VELR consistently achieves the best performance, demonstrating that the ensemble LRM architecture is essential for modeling long-term consistency. These discussions have been added to Tab. 5, Appendix C.7.
>
>
> **References**
>
> [1] Chen et al. Panda-70m: Captioning 70m videos with multiple cross-modality teachers. CVPR 2024.

---

### Author Response · Authors · 2025-12-03
**Summary of Manuscript Revisions**

We sincerely thank the reviewers for their thoughtful and constructive feedback. Their comments have greatly improved the clarity and rigor of the manuscript. In response, we have incorporated all suggested revisions and uploaded an updated version. The main changes are as follows in the order they appear in the manuscript:

* We **clarify the contributions of VELR** in Sec. 1. (suggested by R. geYU)
* We provide additional details of the **online alignment mechanism for LRM** in Sec. 4.3. (suggested by R. geYU, R. QQue)
* We provide more **discussion of the comparison between VADER and VELR** in Sec. 5.2. (suggested by R. u2cT)
* We include more qualitative and quatitive results in Sec. 5.6 to demonstrate the **soundness of the truncated mid-step setting**. (suggested by R. geYU, R. QQue, and R. u2cT)
* We add a figure illustrating **LRM prediction error throughout ReFL training** in Appendix C.6. (suggested by R. geYU)
* We conduct an **ablation study of LRM components** (Table 4 & Fig. 9, Appendix C.7). (suggested by R. geYU)
* We include an experiment on the **long-term consistency under different LRM settings** (Table 5, Appendix C.7). (suggested by R. geYU)
* We add an experiment evaluating the **proxy error of LRM on an OOD test set** (Table 6, Appendix C.7). (suggested by R. geYU)
* We provide a **sensitivity analysis of the truncated mid-step setting** in Appendix C.8. (suggested by R. QQue)
* We report the **parameter scales of video RMs and latent RMs** in Appendix C.9. (suggested by R. QQue)

We believe these revisions, together with the detailed clarifications in our rebuttal, fully address the points raised by the reviewers. We greatly appreciate the reviewers’ time and effort in evaluating our work and for the opportunity to further strengthen the manuscript.

---

### Author Response · Authors · 2025-12-03
**General Response**

Dear AC, SAC, PC, and Reviewers:

We sincerely thank all reviewers for their thorough and thoughtful feedback. Each reviewer provided valuable comments that helped us improve the manuscript and we have addressed all concerns in the rebuttal.

We are encouraged that reviewers recognize the proposed VELR as “**a cohesive system to solve this specific problem and works well**” (R. geYU), tackles “**an important topic with well-articulated motivation**” (R. QQue) and is “**technically sound**” (R. u2cT). We are also pleased that our key component latent reward model is considered to “**enable the effective use of powerful, temporal-aware Video RMs that were previously computationally inaccessible**” (R. u2cT) and “**reduce RL training costs, particularly for video generation**” (R. QQue).

We further appreciate the reviewers’ recognition of our experimental setup. As noted, “**Experimental results indicate the effectiveness of the proposed method**” (R. u2cT), and “**quantifiable achievements like reducing memory usage demonstrate the framework's efficiency**” (R. geYU).

In response to the reviewers’ constructive suggestions, we have addressed all the concerns with new experiments, clearer explanations and additional evidence:

- **For supplementary video material**, we have provided comparison videos through an anonymous project website (https://elvr-anonymous.github.io/VELR_anonymous/) (suggested by R. u2cT). Please kindly check out our attractive results.

- **For design choice**, we have added ablation studies to illustrate that the **truncated mid-step setting** is reasonable, robust, and generalizable (suggested by R. geYU, R. QQue, and R. u2cT). We also conducted further in-depth ablations on **individual components of LRM** to show its long-term consistency and low proxy error on an OOD test set (suggested by R.geYU).

- **For experiments**, we have highlighted the purpose of Fig. 3 in the manuscript to avoid possible misunderstanding (by R. QQue) and to clarify that Fig. 3 has already included the comparison against image RM-based VADER (suggested by R. u2cT). We have added qualitative comparisons to baseline Dollar (suggested by R. QQue).

- **For clarity**, we have reported model sizes of Video RMs and more implementation details of LRM (suggested by R. QQue). We have polished the manuscript to better highlight training reliability (suggested by R. geYU and R. QQue).

VELR introduces an efficient and scalable ReFL framework that achieves up to **87.6% memory reduction** (please see Tab. 1, Sec. 5.1), enabling the application of large-scale video RMs that were previously computationally infeasible and receives a **remarkable preference rate gain (41% vs. 8% on average)** in the human evaluation (please see Fig.6, Sec. 5.5). We believe these contributions will have meaningful impact on future video generative models and RLHF research.

Finally, we appreciate that reviewer u2cT “**will be glad to raise the score rate if the concerns are solved**”. Given that all raised concerns have been thoroughly addressed with new evidence and clarifications, we sincerely hope that the AC can take this into account when making the final decision.

Best regards,

Authors of submission #6523

---

### Note · Authors · 2026-01-27

I have read and agree with the venue's withdrawal policy on behalf of myself and my co-authors.

---

### Meta-Review · Area_Chair_Bnxo · 2026-01-07

**Summary:**

The paper presents VELR, a framework aimed at improving the efficiency of large-scale video reward models (RMs) used in Reward Feedback Learning (ReFL) for text-to-video generation. VELR leverages an ensemble of Latent Reward Models (LRM), which operate in latent space, bypassing the need for backpropagation through large video RMs.

While VELR presents an important and practical contribution to the field of text-to-video generation by addressing the memory bottleneck associated with large-scale video reward models, several concerns about the novelty, empirical validation, and generalizability of the proposed framework prevent this paper from fully meeting the acceptance criteria. Additionally, the lack of strong visual evidence and comparative analysis with state-of-the-art methods significantly weakens the paper's empirical contribution.

However, I still believe the paper is well-written and presents a noteworthy solution with significant potential for reducing the computational cost of large-scale video RMs. I encourage the authors to make further efforts to adequately address these issues and consider submitting to a future venue.

**Reviewer Concerns:**

There are remaining concerns that still need further attention include:
- The potential impact of proxy errors between the ensemble LRM and the ground-truth vRM, especially on out-of-distribution test sets, requires more rigorous quantitative analysis. This includes ensuring that these errors are not amplified during training and negatively affecting model stability.
- While intuitive, the truncated mid-step strategy lacks comprehensive ablation studies to confirm its optimality and generalizability across different models. More evidence is needed to demonstrate that this method consistently outperforms alternatives in various settings.
- While the paper addresses memory efficiency, the trade-off between video quality and efficiency needs more emphasis. Visual comparisons with high-fidelity image reward models and video reward models in the same contexts should be more thoroughly explored to ensure the proposed framework maintains high-quality video generation.
- The choice of the truncated mid-step setting is model-specific, and more work is needed to confirm that this approach can be universally applied to other diffusion architectures without significant loss in performance.

**Reviewer Scores:**

The majority of reviewers acknowledge the paper’s contribution but highlight concerns regarding methodological novelty, proxy risks, and video quality comparisons, leading to likely unchanged ratings (2 negative, 1 positive).

---

### Decision · Program_Chairs · 2026-01-26

Reject